# Antibody Profile of Systemic Sclerosis and Mixed Connective Tissue Disease and Its Relationship with Lung Fibrosis and Pulmonary Hypertension

**DOI:** 10.3390/ijms26125684

**Published:** 2025-06-13

**Authors:** Karolina Niklas, Dorota Sikorska, Tatiana Mularek-Kubzdela, Joanna Witoszyńska-Sobkowiak, Iwona Żychowska, Włodzimierz Samborski

**Affiliations:** 1Department of Rheumatology and Rehabilitation and Internal Diseases, Poznan University of Medical Science, 28 Czerwca 1956r. 135/147, 61-545 Poznan, Poland; dorotasikorska@ump.edu.pl (D.S.); jwitoszynska@orsk.pl (J.W.-S.); izychowska@ump.edu.pl (I.Ż.); wlodzimierz.samborski@ump.edu.pl (W.S.); 2Department of Cardiology, Poznan University of Medical Sciences, Długa 1/2, 60-848 Poznan, Poland; tatianamularek@wp.pl

**Keywords:** antibody profile, lung fibrosis, mixed connective tissue disease, pulmonary hypertension, systemic sclerosis

## Abstract

The most serious complications of systemic sclerosis (SSc) and mixed connective tissue disease (MCTD) include lung fibrosis (LF) and pulmonary hypertension (PH). The aim of this study was to find any association between the serological profile and the incidence of these complications. The tested group included 121 persons (87 SSc, 34 MCTD); mean age 55.6 ± 13.4 years. Patients were qualified for the LF presence group based on HRCT. Likelihood of PH was determined using echocardiography. The presence of antinuclear antibodies (ANA) was assessed using indirect immunofluorescence, ANA-profile, sclerosis-profile (using EUROIMMUN kits), and antiphospholipid antibodies (aPL) (using the ELISA method). Distribution of individual antibody types was at a level similar to the previously described groups in the Polish population and differed from the American and African population. A positive correlation was found between LF and the presence of anti-Scl-70 (*p* = 0.024) antibodies, negative correlation was found between LF and the presence of anti-histone (*p* = 0.03), anti-centromere A (*p* = 0.009), anti-centromere B (*p* = 0.014), and anti-nucleosomes (*p* = 0.03) antibodies. No correlation between the presence of aPL and the above complications was found. The prevalence of individual antibody types in SSc and MCTD may have ethnic and geographical grounds. Scl-70 antibodies correlate positively with LF. Anti-centromere, anti-histone, and anti-nucleosome antibodies reduce its risk. No correlation between aPL and the occurrence of LF and elevated PH risk was found.

## 1. Introduction

Systemic sclerosis (SSc) and mixed connective tissue disease (MCTD) are included in connective tissue diseases (CTD), which can result in a variable clinical picture—from skin and joint lesions to the involvement of internal organs. The most serious complications of these diseases include lung fibrosis (LF) and pulmonary hypertension (PH). These complications can cause considerable deterioration of the quality of life and significantly increase mortality among patients with SSc and MCTD [1]. However, numerous attempts have been made to identify predictors of organ complications in patients with CTD. One of these is the association of serological profile with the clinical course of the disease. In over 90% of the patients with SSc and MCTD, the presence of antinuclear antibodies (ANA) has been detected [2]. ANA can react with various intracellular antigens, such as double-stranded DNA (dsDNA), histones, or extractable nuclear antigens (ENA). The presence of individual types of antibodies is seen as a potential marker of organ complications in CTD. Some authors emphasize that the ethnicity and geographic location of the patient can be of significance, as differences in the occurrence of antibodies and their associated organ complications have been demonstrated depending on these conditions [2,3,4]. The serological profile of the patient may be linked not only to the occurrence of organ complications but also to the long-term survival of those patients who have already been diagnosed with such complications [5]. Another issue is the association of antibodies other than ANA, which are present in case of CTD, with the risk of organ complications. The most commonly discussed of these include antiphospholipid (aPL) antibodies [6].

Therefore, in this study we aimed to determine the antibody profile occurring in SSc and MCTD among patients hospitalized at the Department of Rheumatology and Internal Disease of the Poznan University of Medical Sciences from 2005 to 2017 and the correlation of the presence of individual ANA and aPL antibodies with the incidence of LF and PH.

## 2. Results

In this study, we analyzed the data of 121 patients with SSc (*n* = 87) and MCTD (*n* = 34). The mean age of the experimental group was 55.56 ± 13.44 years (Table 1).

Although a significant age difference was observed between patients with SSc and MCTD, the duration of disease did not differ significantly between both groups, which is a more important issue when it comes to organ complications. We also did not observe any significant difference in the duration of the disease in patients with elevated risk of PH and low risk of PH (*p* = 0.17) and between patients with and without LF (*p* = 0.14).

Of our 121 patients, only two had atrial fibrillation, six had diabetes, and one was obese. Therefore, we considered that these comorbidities did not affect our results. A total of 38 of our patients had hypertension. However, there was no significant difference in the prevalence of hypertension in patients with increased risk of PH and low risk of PH (*p* = 0.73) and between patients with and without LF (*p* = 0.76).

Furthermore, 114 patients (94.2%) showed the presence of ANA at a titer of 1:160 or higher. Analysis of HRCT results demonstrated the presence of LF in 47 patients (38.8%). There was a statistically significant difference between patients with SSc and MCTD, unfavorable for SSc, which may suggest that organ complications are less common in MCTD.

In order to assess the correlation between the occurrence of LF and the presence of individual antibodies in the ANA profile, we used the data of 120 patients (87 with SSc and 33 with MCTD). Table 2 presents the percentage distribution of antibodies in ANA profile.

According to our results, there was a positive correlation between LF and the presence of Scl-70 and a negative correlation between LF and the presence of antibodies against histones, centromere B, and nucleosomes. To assess the association between the occurrence of LF and the individual antibodies in the sclerosis profile, we used data from 101 patients. Table 3 presents the percentage distribution of antibodies in the sclerosis profile.

According to our results, there was a negative correlation between the occurrence of LF and the presence of antibodies against centromeres A and B. The data of 55 patients were used to test the association between occurrence of LF and presence of aPL. According to the results, aPL was detected in 15.9% of the patients with SSc and in 9.1% of the patients with MCTD. Table 4 shows the percentage distribution of patients with aPL along with the presence of aCL or β2GPI.

Because both types of antibodies were detected in some patients with SSc, the total incidence of aPL in SSc was not found to be equal to the sum of percentages of patients with aCL and β2GPI. We did no correlation between the occurrence of LF and the presence of aCL and β2GPI. Table 5 presents the results with respect to the association of LF with the presence of individual antibodies.

Echocardiographic data, which enabled the assessment of PH probability according to the ESC/ERS guidelines of 2015, was obtained from 83 patients. In all, 16 patients (19.3%) were grouped under the EP group and 67 patients (80.7%) were grouped under the LP group. To test the association of EP with individual antibodies in ANA profile, we analyzed the data obtained from 83 patients. There was no correlation between EP and the presence of any of the antibodies. To test the association of the occurrence of EP with the presence of individual antibodies in the sclerosis profile, we analyzed the data obtained from 75 patients. According to the results, there was no correlation between EP and the presence of any of the antibodies. Next, we used data obtained from 47 patients in order to test the association between EP and aPL antibodies. According to the results, there was no relationship between EP and the presence of aCL and β2GPI antibodies. Table 6 presents detailed results.

## 3. Discussion

CTD are strongly linked to the presence of ANA. According to the literature, only in 5–11% patients with SSc, these antibodies are not found [2,7,8]. In this study, 5.8% of patients were found to be negative for ANA. The most common antibodies in SSc are, depending on its form, antibodies against topoisomerase I (Scl-70) (associated with the diffuse sclerosis form—dSSc) and antibodies against centromeres (ACA) (associated with limited sclerosis form—lSSc). In MCTD, the most characteristic antibodies are antibodies against U1-RNP, which by definition are present in 100% of the patients. The prevalence of individual antibodies determined in the ANA and the sclerosis profile may differ depending on ethnic and geographic conditions. This fact has been documented by Meyer et al., who compared the French and American populations of patients with SSc [3]. They recorded a significantly higher prevalence of Scl-70 in the French population. However, the prevalence of anti-RNA polymerase III was significantly higher in the American population. Rodriguez-Reyna et al. examined the Mexican population with SSc and recorded more frequent presence of ACA in comparison to the Japanese and Afro-American population; more frequent occurrence of Scl-70 in comparison to Caucasian and Afro-American population; and high prevalence of anti-PM-Scl, anti-Ku antibodies, antibodies against nucleosomes, and anti-dsDNA in comparison to other populations and lower frequency of anti-RNA polymerase III [4]. Hamaguchi et al. estimated the frequency of Scl-70 and ACA to be higher in the Japanese population than in the Caucasian and Afro-American populations [2]. These authors estimated the presence of U1-RNP to be low in the examined population, whereas anti-Th⁄To, anti-U3-RNP, and anti-RNAP were present as frequently as in other populations. Chang et al. studied the New Zealand population in comparison to the population obtained by the EUSTAR group (European Scleroderma Trials and Research group); their results demonstrated a more frequent occurrence of ACA and anti-RNA polymerase III [9]. Results of this study (Table 1 and Table 2) were found to be similar to those obtained by Wielosz et al. and Żebryk et al. with respect to the Polish population [10,11]. In the case of MCTD, aside from the anti-U1-RNP that is necessary to be determined, we may find the presence of various antibodies; however, studies on the topic are scarce. Ungprasert et al. described the frequency of their occurrence in the American population [12]. Compared to their study, the results of this study exhibited higher prevalence of anti-SS-A (14% vs. 27.3%), anti-SS-B (0% vs. 9.1%), and anti-dsDNA (3% vs. 24.2%), which indicates certain population differences. With regard to the remaining antibodies, the obtained results were at similar levels.

In this study, aPL antibodies were detected in 15.9% of the patients with SSc and 9.1% of the patients with MCTD. In the case of aCL, antibodies were detected in 6.8% of the patients with SSc. These antibodies were not found in patients with MCTD. Anti-β2GPI antibodies were found in 13.6% of the patients with SSc and 9.1% of the patients with MCTD. Gupta et al. detected the presence of aPL antibodies in 9.7% of the patients with SSc [13]. Rai et al. detected the presence of aPL antibodies in 13.3% of the patients with SSc and 13.3% of the patients with MCTD [14]. Thus, these results are comparable to our study. On the other hand, Touré et al. found they most frequently occurred in patients with SSc among all aPL-determined aCL (in 17.5% of patients) and anti-β2GPI antibodies (in 37.5% of patients) [15]. Their results were drawn from the African population and are considerably higher than the results of this study, which may also confirm the influence of ethnic differences in aPL distribution.

A separate issue is the relationship between the presence of individual types of antibodies and organ complications in the case of CTD. The most common complications include LF and PH.

In this study, we determined a positive correlation between the presence of Scl-70 and LF. Other studies have also obtained similar results [2,3,16,17,18,19,20]. LF in SSc may also have geographic grounds, which has been demonstrated in the study conducted by Meyer et al., who detected a more frequent occurrence of LF in French than in American patients (57% vs. 30%) [3]. Moreover, other researchers demonstrated positive association between the other antibodies in the ANA and sclerosis profiles and LF. For example, Hamaguchi detected an elevated risk of LF in patients with SSc and with the presence of anti-Th/To and anti-U3-RNP antibodies [19]. Grassegger et al. also confirmed a positive correlation between the presence of anti-Th/To and LF [17]. Graf et al. observed a trend for increased frequency of LF in patients with SSc and the presence of anti-Pm/Scl and anti-Th/To antibodies [18]. Furthermore, Hachulla et al. detected an association between the occurrence of anti-U1-RNP antibodies and LF [21]. In our study, we were unable to observe any of the aforementioned associations. However, we found a negative correlation between the occurrence of LF and the presence of antibodies against centromeres A and B, histones, and nucleosomes. Our results partially corroborate the results obtained by Cozzani et al., Iniesta Arandia et al., and Wang et al. who described the protective role of ACA relative to the occurrence of LF in patients with SSc [20,22,23]. Thus, we can derive a fresh conclusion regarding the protective role of antibodies against histones and nucleosomes in the development of LF. Morozzi et al. presented only the absence of relationship of anti-histone antibodies with organ complications in SSc [24]. Alaya et al. described the case of a patient with SSc and antibodies against nucleosomes who developed LF. However, this was a case of SSc associated with professional exposure to silica, and LF was probably linked with silicosis than with the presence of antibodies [25].

In this study, we did not find a correlation between LF and the presence of aCL and β2GPI antibodies. Similarly, Touré et al. were unable to find a relationship between the presence of aPL and any organ complication in SSc [15]. However, Morrisroe et al. demonstrated that LF may be linked to the presence of aCL [26], which may be explained by thrombosis of small vessels, leading to various organ symptoms in SSc, including LF.

Another highly important complication in SSc and MCTD is PH [27]. In this study, we were unable to determine any positive correlation between the presence of specific antibodies in the ANA and sclerosis profile and EP of PH as assessed by echocardiographic examination. Earlier publications described such associations, particularly with reference to ACA [17,19,28,29]. Several studies provided reports on the association between the antibodies against U1-RNP and increased probability of PH [16,18,21]. Sobanski et al. compared patients with PH in the course of SSc with and without the presence of antibodies against U1-RNP [5]. The hemodynamic parameters were found to be similar in both groups, yet the presence of antibodies against U1-RNP appeared to play a protective role against mortality due to PH in SSc and other CTD. The presence of anti-Th/To and anti-U3-RNP antibodies increased the risk of PH, which has also been mentioned among other described antibodies [17,18,19].

Among our patients, we have been unable to determine a correlation between elevated risk of PH and the presence of aCL and anti-β2GPI antibodies. Previous publications have described the absence of such associations [13,15]. However, some studies have reported such correlation, both in patients with SSc and MCTD [6,25,30]. Differences may stem from clinical symptoms exhibited by the patients, i.e., whether they demonstrate symptomatic antiphospholipid syndrome. In this study, none of the patients with PH were found to be associated with thromboembolic disease [31].

Antibodies in the diagnosis of SSc and MCTD are routinely determined. Often, in addition to the most typical ones such as anti-Scl-70, anticentromere antibodies, or anti-U1-RNP, we also observe the presence of other antibodies. But they can be useful not only in diagnostics. As shown above, some of them can have a significant relationship with the occurrence of organ complications. Therefore, their detection should oblige us to increase vigilance in the detection of these complications and early implementation of treatment if such complications occur.

## 4. Materials and Methods

In this study, data were obtained based on the retrospective analysis of the medical history of patients hospitalized at the Department of Rheumatology and Internal Disease of the Poznan University of Medical Sciences from 2005 to 2017 (ethical approval was not required—Confirmation KB—656/19). Diagnosis was made or verified in the case of SSc based on the ACR/EULAR criteria from 2013 [32], and in the case of MCTD, based on the criteria by Alarcón-Segovia and Villareal and/or Kasukawa [33,34]. The experimental group included 121 patients: 87 diagnosed with SSc and 34 diagnosed with MCTD.

Data were obtained from the patients with detailed medical history, physical examination, laboratory assessments with special emphasis on serological examinations (ANA and aPL), results of high-resolution computer tomography (HRCT) of the lungs, and results of echocardiography as well as right heart catheterization (RHC), if such had been conducted.

All 121 patients underwent the test to detect the presence of ANA and their titer as well as HRCT. For individual ANA, the so-called ANA profile was determined for 120 patients (87 with SSc and 33 with MCTD). The presence of ANA characteristic for SSc, the so-called sclerosis profile, was determined for 101 patients (85 with SSc and 16 with MCTD). Our collected material dates back to 2005. The missing sclerosis profile data are patients from the earliest years. At that time, the sclerosis profile was not yet determined because we did not have the appropriate assay kits. Therefore, the gaps were not caused by clinicians deciding in whom the sclerosis profile should be determined, but by the inability to perform the sclerosis profile at that time. This is the only reason these patients were excluded from the calculations. Presence of aPL was determined in 55 patients (44 with SSc and 11 with MCTD). Echocardiographic examinations provided data enabling the assessment of likelihood of PH according to 2015 ESC/ERS guidelines [35], and they were finally estimated in 83 patients (69 with SSc and 14 with MCTD). And again—the material we have collected dates back to 2005. Echocardiograms incomplete with respect to the 2015 ESC/ERS criteria come from the earliest years. At that time, significantly fewer parameters were assessed (these studies did not include, for example, the eccentricity index or the right atrial area). This was in no way related to the physician’s decision as to who should have a “more accurate” echocardiogram and who should not—all echocardiographic studies throughout the assessed period were performed by the same experienced cardiologist. So, this is the only echocardiography standard to change over the years. Depending on the value of maximum tricuspid valve regurgitation and presence of other echocardiographic indicators of PH, the patients were divided into three groups: low, moderate, and high risk of PH, and in practice, due to the low numbers of subgroups of moderate and high likelihood of PH, for subsequent calculations, these groups were pooled as a single group, i.e., elevated PH probability (elevated probability (EP), 16 patients) versus the group with low probability of PH incidence (low probability (LP), 67 patients). Most patients with EP also had RHC performed, which is the gold standard for diagnosing PH. The methodology for echocardiography has been described in detail in our previous paper [31].

ANA was detected using indirect immunofluorescence technique; the result was considered to be positive if the antibodies are detected in the serum dilution of 1:160 (titer of 1:160) and higher. ANA and sclerosis profiles were determined using EUROIMMUN kits with automatic assessment of stripes using the EUROLineScan software v. 3.4. Within individual determinations, the presence of antibodies against the following antigens was assessed: in the ANA profile—U1 ribonucleoprotein (U1-RNP), Sm, SS-A, Ro-52, SS-B, Scl-70, PM-Scl, Jo-1, centromere B, PCNA, dsDNA, nucleosomes, histones, ribosomal P protein, AMA-M2; and in the sclerosis profile—Scl-70, centromeres A and B, RNA 11 kDa and 155 kDa, fibrillarin, NOR90, Th/To, PM-Scl100, PM-Scl75, Ku, PDGFR, R0-52. Among the aPL antibodies, the presence of anticardiolipin (aCL) and anti-β-2-glycoprotein-I (β2GPI) antibodies were determined. The tests were performed using ELISA (polyvalent test, detecting antibodies in IgG, IgM, and IgA classes).

HRCT was performed in all 121 patients using a Siemens Emotions 16-slice device. Initially, we intended to refer to ILD in our work, but after analyzing the data, it turned out that out of 47 patients with lung lesions in HRCT, 43 had interstitial fibrosis described, and 4 had honeycomb. In seven patients, changes of a “milk glass” nature were described, but they were always accompanied by fibrosis. Moreover, only in six patients with fibrosis was traction bronchiectasis described at the same time, so we did not distinguish them as a separate criterion. Finally, in view of such results, we decided to focus only on lung fibrosis.

Transthoracic echocardiography was performed using a GE Vivid 7 device (General Electric Healthcare Technologies, Inc., Chicago, IL, USA) with a 3.5 MHz head at the Department of Cardiology of the Poznan University of Medical Sciences.

### Statistical Analysis

Age is presented as the arithmetic mean and standard deviation. The prevalence of individual types of antibodies is presented as the percentage of patients who show the presence of the given antibody relative to the number of patients in whom the given antibody was performed. The relationship between individual antibodies and the occurrence of LF and elevated risk of PH was calculated using Fisher’s exact probability test. A *p* value < 0.05 was considered statistically significant. CSS Statistica v.12.5 software package was used to perform calculations.

The main study limitation is its retrospective character. That is why it was not possible to fill in the gaps that occurred in the documentation. For example, we obtained 120 ANA profiles and 101 sclerosis profiles, which is still a large group from a single center, but with only 55 antiphospholipid antibody results. Nevertheless, we considered it worthwhile to use all available data in our study, although some calculations refer to smaller groups.

For this reason, we could also use only 83 results of echocardiography for our calculation—we had to exclude 38 patients because of incomplete results for the 2015 ESC/ERS criteria. Even fewer patients underwent right heart catheterization (15). So, we decided to base our calculations on the probability of pulmonary hypertension resulting from echocardiographic examination. The strength of our project was the fact that all echocardiographic studies throughout the assessed period were performed by the same experienced cardiologist.

HRCT was performed in all analyzed cases. We focused specifically on pulmonary fibrosis. We did not include non-fibrotic interstitial lung disease.

## 5. Conclusions

The frequency of incidence of individual types of antibodies in SSc and MCTD may have ethnic and geographic grounds. Scl-70 antibodies are associated positively with LF, whereas ACA, anti-histone, and anti-nucleosome antibodies reduce the risk of LF. There was no influence of any of the antibodies from ANA and sclerosis profile on the increased risk of PH. We did not find an association between aPL with the occurrence of LF and elevated risk of PH.

## Figures and Tables

**Table 1 ijms-26-05684-t001:** Characteristics of the studied group.

	Whole Group *n* = 121	SSc *n* = 87	MCTD *n* = 34	SSc vs. MCTD
Age (years)	55.56 ± 13.44	57.81 ± 12.78	49.79 ± 13.53	*p* = 0.0028
Sex (F/M)	108/13	77/10	31/3	*p* = 0.67
Duration of illness (years)	12.07 ± 8.26	11.87 ± 8.79	12.56 ± 6.81	*p* = 0.68
Current smoking (yes/no)	15/106	14/73	1/33	*p* = 0.06
Arterial Hypertension (yes/no)	38/83	31/56	7/27	*p* = 0.12
Positive family history of rheumatic diseases (yes/no)	20/101	15/72	5/29	*p* = 0.74
LF presence (yes/no)	47/74	42/45	5/29	*p* = 0.027
EP/LP of PH ^a^	16/67	15/54	1/13	*p* = 0.22

^a^ It affects only patients with complete echocardiographic data, so the number *n* in individual subgroups is less than for other parameters. LF—lung fibrosis; PH—pulmonary hypertension; EP—elevated probability of PH; LP—low probability of PH.

**Table 2 ijms-26-05684-t002:** Percentage distribution of the individual antibodies in ANA profile.

	SSc (*n* = 87)	MCTD (*n* = 33)
U1-RNP	5.7%	100%
Sm	3.4%	21.2%
SS-A	6.9%	27.3%
Ro-52	25.3%	51.5%
SS-B	4.6%	9.1%
Scl-70	41.4%	6.1%
PM-Scl	11.5%	12.1%
Jo-1	0%	0%
Centromere B	27.6%	0%
PCNA	2.3%	6.1%
dsDNA	9.2%	24.2%
Nucleosomes	1.1%	18.2%
Histones	0%	21.2%
Ribosomal P protein	1.1%	6.1%
AMA-M2	8.0%	3.0%

Abbreviations: ANA, antinuclear antibody; SSc, systemic sclerosis; MCTD, mixed connective tissue disease; U1-RNP, U1 ribonucleoprotein.

**Table 3 ijms-26-05684-t003:** Percentage distribution of the individual antibodies in the sclerosis profile.

	SSc (*n* = 84)	MCTD (*n* = 17)
Scl-70	40.5%	5.9%
Centromere A	22.6%	11.8%
Centromere B	27.4%	0%
RNA 11 kDa	3.6%	0%
RNA 155 kDa	3.6%	5.9%
Fibrillarin	8.3%	17.6%
NOR90	0%	5.9%
Th/To	4.8%	11.8%
PM-Scl100	13.1%	17.6%
PM-Scl75	19.0%	17.6%
Ku	4.8%	11.8%
PDGFR	2.4%	0%
R0-52	27.4%	47.1%

Abbreviations: SSc, systemic sclerosis; MCTD, mixed connective tissue disease.

**Table 4 ijms-26-05684-t004:** Percentage distribution of patients with different types of aPL antibodies.

	SSc (*n* = 44)	MCTD (*n* = 11)
aCL	6.8%	0%
β2GPI	13.6%	9.1%

Abbreviations: aCL, anticardiolipin; aPL, antiphospholipid; β2GPI, anti-β-2-glycoprotein-I; SSc, systemic sclerosis; MCTD, mixed connective tissue disease.

**Table 5 ijms-26-05684-t005:** Association between LF and the presence of individual antibodies in the ANA profile, sclerosis profile, and aPL.

ANA-Profile (Yes/No) *n* = 120	LF Present	LF Absent	*p*
U1-RNP	8/38	30/44	*p* = 0.06
Sm	2/44	8/66	*p* = 0.18
SS-A	5/41	10/64	*p* = 0.45
Ro-52	13/33	26/48	*p* = 0.28
SS-B	1/45	6/68	*p* = 0.17
Scl-70	20/26	18/56	*p* = 0.024
PM-Scl	7/39	7/67	*p* = 0.25
Jo-1	0/46	0/74	
Centromere B	4/42	20/54	*p* = 0.01
PCNA	2/44	2/72	*p* = 0.50
dsDNA	5/41	11/63	*p* = 0.37
Nucleosomes	0/46	7/67	*p* = 0.03
Histones	0/46	7/67	*p* = 0.03
Ribosomal P protein	0/46	3/71	*p* = 0.23
AMA-M2	1/45	7/67	*p* = 0.116
sclerosis profile (yes/no) *n* = 101			
Scl-70	19/25	16/41	*p* = 0.085
Centromere A	4/40	17/40	*p* = 0.009
Centromere B	5/39	18/39	*p* = 0.014
RNA 11 kDa	2/42	1/56	*p* = 0.403
RNA 155 kDa	1/43	3/54	*p* = 0.412
Fibrillarin	3/41	7/50	*p* = 0.286
NOR90	0/44	1/56	*p* = 0.564
Th/To	4/40	2/55	*p* = 0.225
PM-Scl100	8/36	6/51	*p* = 0.207
PM-Scl75	12/32	7/50	*p* = 0.051
Ku	2/42	4/53	*p* = 0.469
PDGFR	0/44	2/55	*p* = 0.316
Ro-52	11/33	20/37	*p* = 0.192
aPL (yes/no) *n* = 55			
aCL	2/26	2/25	*p* = 0.681
β2GPI	3/25	4/23	*p* = 0.479

Abbreviations: aCL, anticardiolipin; aPL, antiphospholipid; β2GPI, anti-β-2-glycoprotein-I; ANA, antinuclear antibody; LF, lung fibrosis; U1-RNP, U1 ribonucleoprotein.

**Table 6 ijms-26-05684-t006:** Relationship of EP with the presence of individual antibodies in the ANA profile, sclerosis profile, and aPL.

ANA-Profile (Yes/No) *n* = 83	EP	LP	*p*
U1-RNP	2/14	17/50	*p* = 0.226
Sm	0/16	7/60	*p* = 0.209
SS-A	1/15	7/60	*p* = 0.518
Ro-52	6/10	20/47	*p* = 0.377
SS-B	0/16	4/63	*p* = 0.417
Scl-70	7/9	27/40	*p* = 0.508
PM-Scl	1/15	10/57	*p* = 0.325
Jo-1	0/16	0/67	
centromere B	1/15	16/51	*p* = 0.105
PCNA	1/15	2/65	*p* = 0.479
dsDNA	0/16	12/55	*p* = 0.062
nucleosomes	0/16	4/63	*p* = 0.417
histones	0/16	4/63	*p* = 0.417
ribosomal P protein	0/16	1/66	*p* = 0.807
AMA-M2	1/15	3/64	*p* = 0.583
sclerosis profile (yes/no) *n* = 75			
Scl-70	7/9	24/35	*p* = 0.522
centromere A	2/14	13/46	*p* = 0.324
centromere B	2/14	15/44	*p* = 0.300
RNA 11 kDa	0/16	1/58	*p* = 0.787
RNA 155 kDa	0/16	3/56	*p* = 0.481
fibrillarin	2/14	6/53	*p* = 0.545
NOR90	0/16	0/59	
Th/To	1/15	2/57	*p* = 0.519
PM-Scl100	1/15	10/49	*p* = 0.262
PM-Scl75	1/15	10/49	*p* = 0.262
Ku	1/15	4/55	*p* = 0.712
PDGFR	0/16	2/57	*p* = 0.617
Ro-52	6/10	19/40	*p* = 0.453
aPL (yes/no) *n* = 47			
aCL	0/10	4/33	*p* = 0.370
β2GPI	1/9	5/32	*p* = 0.622

Abbreviations: aCL, anticardiolipin; aPL, antiphospholipid; β2GPI, anti-β-2-glycoprotein-I; ANA, antinuclear antibody; EP, elevated probability; U1-RNP, U1 ribonucleoprotein.

## Data Availability

The raw data supporting the conclusions of this article will be made available by the authors on request.

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
