# Peer review of "Antibody Profile of Systemic Sclerosis and Mixed Connective Tissue Disease and Its Relationship with Lung Fibrosis and Pulmonary Hypertension"

_ijms, 2025, doi:10.3390/ijms26125684_

Round 1

Reviewer 1 Report (New Reviewer)

Comments and Suggestions for Authors I read the paper with interes. Although there have been several reports on the relationship between autoantibodies and phenotypes in systemic sclerosis and MCTD, as the authors note that this may vary by region and race, there are still many unknowns, so I think this is a important paper with clinical situation.   I thought that the use of echocardiographic data to evaluate the presence or absence of pulmonary hypertension was a bit of a limitation as a paper. However, it is understandable that this is actual clinical data with limitations.   The problem is that the statistical method is inappropriate. I think that the results may have changed significantly due to statistics. The chi-square test is based on several approximations and has been reported to be accurate only when all expected numbers are large. It has been pointed out that the analysis results are likely to be inaccurate, especially when 20% or more of the expected numbers are less than 5. In this study, there are a large number of items with expected values ​​less than 5, so it is recommended to use Fisher's exact probability test instead of the chi-square test.   The following is only comment of my own opinion. Of course, the content is clinically important and I believe it provides useful information for readers. However, it only describes the observed facts descriptively, the number of cases is not so large, there is not much data related to clinical significance, and the mechanism is not discussed, so I felt it lacked novelty a bit.
For example, it would be better to evaluate prognostic data, consider the severity of pulmonary fibrosis and pulmonary hypertension, or include pulmonary hypertension with and without pulmonary fibrosis in the analysis. I felt that readers would find it more interesting if it contained data that answered some more clinical questions.

Author Response

Dear Sir or Madam,

Yours faithfully

Karolina Niklas

Reviewer 2 Report (New Reviewer)

Comments and Suggestions for Authors

I appreciated the opportunity to review this retrospective study that investigates the correlation between specific autoantibodies and the presence of lung fibrosis and pulmonary hypertension in patients with systemic sclerosis and mixed connective tissue disease , I just have some suggestions that I think can help improve the paper overall. 

  • - the authors mention that echocardiographic assessments changed over the years (how?). It might help to state whether the interpretation of older data might introduce bias and how did the authors managed  inconsistencies statistically.                                        
  • The authors observed negative association between anti-dsDNA antibodies and PH, i personally found this data novel and intriguing., and i would suggest to back it up with  hypothesized biological mechanisms to explain why this might occur, using literature data

Author Response

Dear Sir or Madam,

Yours faithfully

Karolina Niklas

Round 2

Reviewer 1 Report (New Reviewer)

Comments and Suggestions for Authors

Thank you very much for correcting some points.

However, the response I received was a little disappointing. The biggest problem with this paper is that the statistical method is incorrect. Below is the same text as last time (but written a little shorter).

The problem is that the statistical method is inappropriate. I think that the results may have changed significantly due to statistics. The chi-square test has been reported to be accurate only when all expected numbers are large. So it is recommended to use Fisher's exact probability test instead of the chi-square test in this paper.

The following is a report on the analysis of antibodies in scleroderma using Fisher's exact probability test.

https://pmc.ncbi.nlm.nih.gov/articles/PMC10241190/

The response did not show that the authors understand the serious problems with statistics, so I would like to point out the following point as an example of a problem that may arise from incorrect statistics.

The conclusion of this paper, authors stated as follows:
"Scl-70 antibodies are associated positively with LF, whereas ACA, antihistone, and antinucleosome antibodies reduce the risk of LF. There was no influence of any of the antibodies from ANA and Sclerosis profile on the increased risk of PH, whereas antibodies against dsDNA were found to be exhibit negative relationship with the increased risk of PH. We did not find an association between aPL with the occurrence of LF and elevated risk of PH."

Conclusions may change if statistical methods are changed. Unfortunately, I think there are scientific problems with the current statistical methods.

As for responses other than mentioned above, of course, I believe that if various data were provided, the paper would be more meaningful to readers, but it is understandable that this is a clinical paper and there are limitations to the evaluation items.

Author Response

Dear Sir or Madam,

Yours faithfully

Karolina Niklas

This manuscript is a resubmission of an earlier submission. The following is a list of the peer review reports and author responses from that submission.

Round 1

Reviewer 1 Report

Comments and Suggestions for Authors

Dear authors,

Thank you for the opportunity to review your study, which looked at associations of various serologic parameters of connective tissue disease and their associations with the presence of PH and interstitial lung disease. There are clear practical applications for use of serology in assessing pretest probability for PH and ILD in CTD, particularly as preliminary screening tools for these conditions. There has been significant work previously done in assessing serologic profiles in CTD as risk factors for PAH and ILD, but unfortunately to date we really have no good serologic predictors of ILD and PH in CTD. This study aimed to expand on this work, but unfortunately I believe it has several major limitations, which I detail below.

  1. A study of this sort which significantly impacts clinical practice would have explicit, granular, and complete characterization of the population; i.e., 100% patients would have had all serologic markers of interest checked, ILD is assessed by validated parameters, and PH is defined by right heart catheterization. Any study which does less than that is at risk of bias and confounding by indication; for example, patients who are thought to have ILD are more likely to have markers checked that are associated with ILD (for example, anticentromere antibodies) and patients who are thought to have PH are more likely to undergo echocardiography. In this study, several issues are particularly striking:
    1. A significant proportion of included patients did not have all serologic markers checked. I believe this precludes meaningful determination of association with clinical manifestations of disease.
    2. The definition for ILD in this study is “the presence of fibrosis or lesions with the characteristic ‘milk-glass’ or ‘honeycombing’ appearance.” This definition confuses the idea of fibrotic ILD with non-fibrotic ILD. There are standard definitions for ILD from ATS that are used in research studies, and I would encourage the authors to classify their patients completely with well-validated, standard definitions to maximize the external validity. I am also unclear about how many patients (or whether all patients) underwent HRCT.
    3. The definition of PH in this study is made predominantly by echocardiography. This is really a major limitation of this work. Prior work has consistently demonstrated that echocardiography is inadequate for the diagnosis of pulmonary hypertension, particularly in patients with ILD. A study that aims to associate serologic markers with PH really must have all PH patients characterized by right heart catheterization. Furthermore, a significant proportion of the included population did not have echocardiograms performed, which further raises the questions about association, as noted earlier.
  2. This study is not presented in accordance with STROBE reporting requirements (https://www.equator-network.org/reporting-guidelines/strobe/). Specifically, it is missing a flow diagram, a Table 1 which completely describes the demographics of the study population, etc. The absence of these leads to significant issues with external validity.
  3. I have major concerns about the line “ethical approval was not required” in the Methods section. It is highly irregular that research involving human subjects is not approved by an Institutional Review Board. For obvious practical reasons, this study would not be expected to have a requirement of informed consent, but I would argue that a lack of review of this study by an IRB is an ethical violation of human subjects research.
  4. This study fails to account for patients’ other clinical factors that are associated with the development of PH, including cardiac comorbidities, heart failure, etc., as well as ILD, including duration of disease, types of immunosuppression used, etc. Thus, it cannot meaningfully describe the independent association of any serologic marker with the development of either ILD or PH.
  5. I am concerned that none of these limitations are addressed in the manuscript discussion explicitly.

Comments on the Quality of English Language

The overall quality of English is acceptable, but does merit proofreading for grammatical correctness and clarity.

Author Response

Dear Sir or Madam

Your faithfully

Karolina Niklas

Reviewer 2 Report

Comments and Suggestions for Authors

In this manuscript (ID# ijms-3549639), entitled “Antibody Profile of Systemic Sclerosis and Mixed Connective Tissue Disease and Its Relationship with Lung Fibrosis and Pulmonary Hypertension”, authors Niklas et al studied the antibody profile, including antinuclear and antiphospholipid antibodies, in patients with systemic sclerosis and mixed connective tissue disease. Their results demonstrate that several antibodies are associated with lung fibrosis and systemic sclerosis. While the findings are interesting, similar association studies have been conducted previously. However, given that the data in this study were collected from Poland, it adds a relatively novel perspective. Nevertheless, there are several major concerns, which are outlined in the following paragraphs:

  1. In the context of lung fibrosis, only some patients with idiopathic pulmonary fibrosis may be associated with antinuclear antibodies. The patients included in this study were suffering from idiopathic pulmonary fibrosis or all types of pulmonary fibrosis?
  2. No healthy control group was included in the study. Therefore, it is unclear whether these antibodies are also present in healthy individuals within this population. Up to 30% of healthy individuals may have a positive ANA test.
  3. In this study, were the patients with pulmonary hypertension also suffering from lung fibrosis? It would be more interesting to compare the antibody profile between patients with lung fibrosis and pulmonary hypertension and those with lung fibrosis without pulmonary hypertension to better understand the prognosis of lung fibrosis.
  4. In the Discussion section, please provide a concise and clear conclusion, the clinical significance, and the impact in this field. This information will be more helpful for readers

Comments on the Quality of English Language

This manuscript is difficult to understand. The English language and grammar need improvement.

Author Response

(The authors gave the same response as above.)

Round 2

Reviewer 1 Report

Comments and Suggestions for Authors

Dear authors,

I appreciate the time and effort spent in addressing my queries. As someone who has served as a principal investigator for many retrospective studies, I appreciate the challenges associated with missing data.

However, in my opinion, your paper does not accomplish what it sets out to do; namely, to determine the antibody profile in SSc and MCTD as related to lung fibrosis and pulmonary hypertension. I elaborate below:

1. To reiterate my initial point, you cannot make conclusive statements about incidence or prevalence when you have significant amounts of missing data. In your study, for example, you are missing approximately 19 patients out of 121 in determining a sclerosis profile. Although it is commendable that you have a reasonably large sample size from a single center, when you have 16% of the data missing for a primary aim of your study, your findings become very fragile. Your response that "the number of patients in whom antibodies were performed determined our calculations of antibody prevalence" is flawed, because your definition of prevalence is subject to confounding by indication; i.e., your calculated prevalence may have been inflated by the decisions of the clinicians at the time to check these antibody profiles and thus reflects the prevalence among those patients clinically suspected to have the disease, not among the general population of patients with SSc or MCTD.

2. I appreciate that you made an effort to only collect the highest quality echocardiograms in your data. Unfortunately, like the observation above, when you only have data for 68.5% of patients of a 121-patient study, I do not believe you can draw meaningful conclusions about prevalence of even the probability of pulmonary hypertension in that population. To make my point more clear: if I were a clinician who strongly suspected the presence of pulmonary hypertension in a patient, I would likely ensure that the echocardiogram quality would be very high; I might be less attentive if I were not worried about the presence of pulmonary hypertension. Thus, the prevalence of pulmonary hypertension in my population would, potentially, be much higher than in the general population of SSc or MCTD because I have enriched my high-quality echo population with patients who have high clinical probability of PH. Your observation that 38 patients were excluded because of incomplete echocardiograms suggests significant technical heterogeneity in your center, which is not uncommon, but renders any conclusions about the prevalence of pulmonary hypertension very limited. 

3. In your responses, you did not address my point about clinical factors that confound the development of PH or ILD. For example, you do not describe whether your patients had atrial fibrillation, diabetes mellitus, obesity, or systemic hypertension; all of these increase the risk of HFpEF (and therefore group 2 PH). You do not describe whether your patients were on immunosuppression or how long they have had connective tissue disease; these are factors that clearly affect the development of ILD. You assume, without providing supporting evidence, that your patients were completely evenly matched in terms of these factors.

4. "Lung fibrosis" and ILD are not the same thing. I assume that your use of the term "milk-glass" refers to "ground-glass opacities," as is convention in ATS and ERS documents. If this is the case, then "milk-glass" is not indicative of fibrotic lung disease; it is, in fact, indicative of active, potentially-reversible parenchymal inflammation. If my assumption is incorrect, please define your use of the term "milk-glass." In addition, the findings of traction bronchiectasis would also suggest fibrotic ILD, alongside honeycombing; in fact, this is often more common in the NSIP pattern associated with SSc and MCTD than the UIP pattern that is associated with honeycombing. Please explain why you did not include traction bronchiectasis in your review of HRCT. 

I would like to offer the authors examples of high-quality publications that address a very similar topic:

Sangani RA, et al. Pulm Circ 2022 (https://doi.org/10.1002/pul2.12117)

Launay D, et al. CHEST 2011 (https://doi.org/10.1378/chest.10-2473)

Jiang Y, et al. Autoimmunity Reviews (https://doi.org/10.1016/j.autrev.2020.102602)

Author Response

Dear Sir or Madam,

Your faithfully

Karolina Niklas

Reviewer 2 Report

Comments and Suggestions for Authors

The revised manuscript has been improved. No further recommendation. 

Author Response

Dear Sir or Madam,

Thank you very much for your review, your time and for all your comments and advice for the future.

Your faithfully

Karolina Niklas

Round 3

Reviewer 1 Report

Comments and Suggestions for Authors

.

Author Response

Dear Sir or Madam,

Thank you very much for your review, your time and for all your comments and advice for the future.

The manuscript has been completed.

Your faithfully

Karolina Niklas